# Quantitative Study of the T-AVIM-Based Simulated IMU Error in Polar Regions

**DOI:** 10.3390/s22165988

**Published:** 2022-08-11

**Authors:** Zhe Wen, Hongwei Bian, Rongying Wang, Heng Ma

**Affiliations:** College of Electrical Engineering, PLA Naval University of Engineering, Wuhan 430033, China

**Keywords:** inertial navigation system (INS), polar navigation, transverse coordinate system, inertial measurement unit (IMU), Earth’s sphere model, simulation test

## Abstract

For solving the problem of polar performance of the inertial navigation system (INS) at mid-low latitudes, the simulation test system constructed by the “attitude and velocity invariant method of trajectory transfer rule based on the transverse coordinate system (T-AVIM)” of the Earth sphere model is used. The test system structure, especially the IMU conversion formula from mid-low latitudes to polar region simulation test, is introduced, and it is proved that the IMU conversion error can be equivalently superimposed on the bias error of the polar simulated IMU. According to the marine estimation formula for the effect of the reference error on the IMU conversion error, the specific influence of the constant error component and the random error component of the reference system on the simulated IMU is analyzed. The calculation method of the simulated IMU error is given with examples and intuitively explained, and the correctness of the theory is verified through simulation experiments.

## 1. Introduction

With global warming and sea ice melting in the Arctic Ocean, navigation in the Arctic is becoming more frequent, and it is of great importance for the research, development, and utilization of the Arctic [1,2]. To promote the research of navigation technology and equipment in the polar region, as an important autonomous navigation means, the polar performance verification of INS has become an important procedure. Facing the special conditions of the polar region, INS uses a special polar control mechanism to calculate the navigation parameters [3,4], so it is necessary to check the performance of INS in the polar mode. However, the polar area sea test has a long period and high cost, which is restricted by natural conditions and difficult to implement. Therefore, trying to simulate the polar area’s performance test of INS in mid-low latitudes has become a probable program. Synchronously collecting the original data from the tested INS and the reference system in mid-low latitudes, based on a certain law, for example, the attitude and velocity invariant to the transverse coordinate system method (T-AVIM), to implement trajectory conversion, a certain algorithm is used to estimate and evaluate the polar region’s accuracy performance for the tested INS, which will be covered in details in Section 2. This technology is called “INS polar mid-low latitude simulation test technology” (abbreviated as “INS P-ML simulation test”).

In the current research in the field of INS P-ML simulation tests, the main methods can be divided into the following categories. First, select virtual poles on the surface of the Earth and reconstruction of the latitude and longitude network of the Earth, make the measured trajectory in the virtual polar region, and provide the inertial navigation solution of the wandering azimuth mechanism [5,6], which is called virtual polar technology. However, it still uses the measured original IMU data and cannot simulate the real physical environment of the polar regions. Second, another method of rotating the Earth Centered Earth Fixed (ECEF) coordinate system twice is adopted to transfer the trajectory to the polar region [7], which cannot solve the deformation problem of the surface trajectory since the Earth is approximately an ellipsoid. Third, adopting the trajectory transfer principle of T-AVIM [8,9,10,11] or a grid coordinate system (G-AVIM [12,13]) and using the recursion method of the reference velocity to construct the polar reference trajectory solves the problem of trajectory deformation caused by different surface curvatures. The mathematical relationship between the simulated IMU in the polar region and actual IMU in the mid-low latitudes of this method is also deduced in the above literature. Fourth, using the virtual sphere method [14] to update the polar trajectory reference makes the algorithm more optimized and simpler while retaining the accuracy of the Earth ellipsoid model. Finally, it is verified that the performance of the simulated INS in the polar region is equivalent to the field test in the polar region under an ideal reference system [8].

It is a feasible way to build a simulation test system using the principle of trajectory transfer with T-AVIM. In the simulation test system, to obtain simulated IMU data at the corresponding trajectory of the polar region, real-time measured reference navigation parameters are required for the conversion of real IMU data. Therefore, the accuracy of the reference systems is vital to simulated IMU data. Previous research has made theoretical derivation, and the calculation formula of the IMU conversion error expressed by the reference error has been studied [15]. Based on the IMU conversion error formula, a quantitative analysis should be further studied, especially for the propagation form of IMU conversion error in the simulated INS and the way simulated INS solution can be influenced.

The rest of this paper is organized as follows: In Section 2, the basic theory of the INS P-ML simulation test method and its system structure is further introduced, together with the IMU conversion formula and its error calculation formula are given. The definition and basic parameters of the transverse coordinate system (refer to [16,17,18,19], etc.) are omitted due to the compactness of the article. In Section 3, the propagation rules for IMU conversion error are studied in theoretical derivation. Some examples are given to describe the quantitative calculation process of constant and random additional biases for simulated IMU. In Section 4, tests are designed to simulate the reference constant (installation) error and random errors, and the theory and formula proposed are verified. Finally, Section 5 sums up the whole paper’s contents, induces relevant conclusions, and analyzes the value of the research.

## 2. Basic Theory of INS P-ML Simulation Test

### 2.1. Main Content and Analysis of Simulation Test Method

The research on the polar mid-low latitude simulation test is to find a method to obtain the polar performance of the inertial navigation system by simulating the polar trajectory and polar IMU data using the information from the mid-low latitude real test. The specific methods and processes are as follows:

Step 1: Mid-low latitude experiments.

Carry out the dynamic inertial navigation test of the carrier in the mid-low latitude areas and simultaneously collect the full navigation reference information (position, velocity, and attitude) of the benchmark equipment and the gyro and accelerometer output of the tested inertial navigation IMU;

Step 2: Calculation of reference (benchmark) information for polar simulation experiments based on the reference of mid-low latitude experiments.

Calculate the reference information (position, velocity, and attitude) for the same maneuvering and navigation process of the mid-low latitudes test in the simulated polar region using certain transfer criteria (T-AVIM used in this paper). It is necessary to obtain the same navigation state test track in the high-latitude region (artificially set). Finally, full simulated reference navigation parameters (position, velocity, and attitude) represented by transverse coordinates are derived if T-AVIM is taken. This step is also called “reference transfer” or “trajectory transfer”;

Step 3: The calculation of the polar simulated inertial navigation IMU data based on the mid-low latitude tested IMU data and the reference information.

The data of IMU (gyro and accelerometer) of the polar simulated INS is calculated according to a certain mathematical conversion compensation algorithm to approximate the actual IMU measurement data output in the simulated polar region track (obtained by the previous step). Usually, the actual tested IMU data and the reference information (including polar simulated reference and actual reference from mid-low latitudes) are both used. This step is also called “IMU data conversion”.

Step 4: Solving for the inertial navigation parameters in the simulated polar region.

The polar region measurement output of the IMU gyro and accelerometer generated by simulation is input to the inertial navigation system of the tested INS, and the polar mechanism (e.g., transverse mechanism for T-AVIM) is taken to solve navigation parameters.

Step 5: Error analysis and evaluation of the polar performance by tested INS.

Under the unified polar coordinate system, the error between navigation parameters solved by polar simulated INS (obtained in Step 4) and the simulated reference navigation parameters (obtained in Step 2) will be compared and calculated, and the accuracy of the tested INS in polar regions is analyzed and evaluated.

From a technical point of view, the scheme has specific advantages and characteristics because it uses the actual IMU data to generate the simulated IMU data. Thus, the generated data contains certain dynamic error characteristics of the tested inertial navigation IMU under the maneuvering state, which is authentic.

It is realized that the key to this method lies in the accuracy of the IMU data conversion calculation. Once the conversion error of the simulated IMU data is large, there will consequently be a large error in the polar solution of the tested INS, and it is difficult to correctly reflect the tested inertial navigation performance. The effectiveness of the evaluation process will decrease significantly or even lose its meaning. Therefore, IMU data conversion errors must be controlled.

Tracing back to the origin of IMU data conversion, the largest source of error comes from the reference system error. Thus, the reference system error should be studied for its influence on simulated IMU data, and the research is further conducted in this paper.

Intuitively, a larger reference error will bring a larger IMU data conversion error, which is not tolerant for higher accuracy INS simulation tests. Therefore, one of the significances of related research is to match the appropriate reference system for the tested INS with different precision levels.

### 2.2. T-AVIM Based Simulation Test System under Earth Sphere Model

It is required to pursue high precision in actual applications. While the estimation of IMU data conversion error is used for reference system selection, the accuracy requirement of error analysis is not urgent. To seek convenience and effectiveness of the research, the Earth sphere model is used to simplify the formula of trajectory transfer and IMU data conversion because there will be complexities when the ellipsoid model is used.

For the structure of the INS P-ML simulation test system using T-AVIM under the Earth sphere model, here is a brief description in Figure 1, where “AVP” is the abbreviation of navigation parameters of attitude, velocity, and position. Reference transfer and IMU data conversion are carried out at the same time once the trajectory transition angle Λ is set.

Assuming that the carrier with the reference system is moving in the mid-low latitudes carrying the strap-down IMU to be tested (a reasonable definition of the transverse coordinate system makes the actual trajectory near the transverse equator), the trajectory navigation parameters of the reference are projected from the traditional geographic frame (*g*-frame) to the transverse geographic coordinate system (*t*-frame) and expressed as transverse latitude $\varphi^{t}$ and longitude $\lambda^{t}$, transverse velocity $v^{t}$, and transverse attitude cosine matrix $C_{b}^{t}$. Rotate the reference trajectory of the measured carrier along the transverse latitude of the surface to the polar region to obtain the simulated reference trajectory, and record the simulated reference trajectory as $\left(\varphi_{m}^{t},\lambda_{m}^{t}\right)$:(1)$$\left(\varphi_{m}^{t},\lambda_{m}^{t})=(\varphi^{t},\lambda^{t}-\Lambda \right)$$
where Λ denotes the trajectory transition angle, which is artificially designed, the complete reference parameters in the simulated polar region are derived (A, V remain unchanged). Figure 2 is a simple diagram of the transfer of the reference track.

The measured IMU data is converted into simulated IMU data through IMU data conversion.

Let “*t_m_*” represent the transverse geographic coordinate system of the simulated polar area and “*b_m_*” represent the body frame of the carrier in the simulated polar area. According to T-AVIM of trajectory transfer, the theoretical relationship between the polar simulated IMU data accelerometer $f_{ib_{m}}^{b_{m}}$, gyroscope $\omega_{ib_{m}}^{b_{m}}$ and the real IMU data in test $f_{ib}^{b}$, $\omega_{ib}^{b}$ is as follows:(2)$$\left\{\begin{array}{l} f_{ib_{m}}^{b_{m}}=f_{ib}^{b}+df^{b}\\ \omega_{ib_{m}}^{b_{m}}=\omega_{ib}^{b}+d\omega^{b} \end{array}\right.$$
where $df^{b}$, $d\omega^{b}$ is called IMU data conversion variables of accelerometer and gyroscope. In the mid-low latitudes and the simulated polar regions, the equations can be written separately to compare, and the results will be derived:(3)$$\left\{\begin{array}{l} df^{b}=-C_{t}^{b}\left\{\left[C_{e}^{t}\left(2\omega_{ie}^{e}+\omega_{et}^{e}\right)-C_{t}^{t_{m}}C_{e}^{t}\left(2\omega_{ie}^{e}+\omega_{et_{m}}^{e}\right)\right]\times v^{t}+g^{t_{m}}-g^{t}\right\}\\ d\omega^{b}=-C_{t}^{b}(\omega_{ie}^{t}-\omega_{ie}^{t_{m}}+\omega_{et}^{t}-\omega_{et_{m}}^{t_{m}}) \end{array}\right.$$
where $g^{t}$, $g^{t_{m}}$ denotes the gravity vector of the actual and simulated area. It is seen that all the variables in the formula are required for the reference parameters; obviously, the reference error will cause the calculation error of IMU conversion variables and then affect the simulated IMU data and the test ability of the simulation test system.

### 2.3. IMU Data Conversion Error

The reference speed and attitude measurement model are expressed in the *t*-frame as (4):(4)$$\left\{\begin{array}{l} {\tilde{v}}^{t}=v^{t}+\Delta v^{t}\\ {\tilde{C}}_{b}^{t}=[I-(\Phi^{t}\times )]C_{b}^{t} \end{array}\right.$$
where “~” means the measured value by the reference system, $\Phi^{t}={\left[\phi_{\mathrm{tE}}^{t},\phi_{\mathrm{tN}}^{t},\phi_{\mathrm{tU}}^{t}\right]}^{T}$ indicates the transverse attitude misalignment angle error, “×” means the skew-symmetric matrix for a 3-d vector. Only marine navigation is considered, so $\Delta v^{t}={\left[\Delta v_{\mathrm{tE}}^{t},\Delta v_{\mathrm{tN}}^{t},0\right]}^{T}$. The reference positioning accuracy is relatively high, and its influence on the process of IMU conversion can be ignored.

Substituting the reference speed and attitude measurement model into Formula (3) of the IMU conversion variables and subtracting it from the theoretical ideal value derives the relationship between the reference error and the calculation error of the IMU conversion variables [15].
(5)$$\left\{\begin{array}{l} \Delta (df^{b})\approx C_{t}^{b}M_{1}\times \Delta v^{t}+\Delta C_{t}^{b}(M_{1}\times v^{t}-G_{t,t_{m}})\\ \Delta (d\omega^{b})\approx \Delta C_{t}^{b}M_{2} \end{array}\right.$$
where
(6)$$\left\{\begin{array}{l} M_{1}=C_{t}^{t_{m}}C_{e}^{t}\left(2\omega_{ie}^{e}+\omega_{et_{m}}^{e}\right)-C_{e}^{t}\left(2\omega_{ie}^{e}+\omega_{et}^{e}\right)\\ M_{2}=C_{t}^{t_{m}}C_{e}^{t}\left(\omega_{ie}^{e}+\omega_{et_{m}}^{e}\right)-C_{e}^{t}\left(\omega_{ie}^{e}+\omega_{et}^{e}\right)\\ G_{t,t_{m}}=g^{t_{m}}-g^{t}={\left[0,0,g^{t}-g^{t_{m}}\right]}^{T} \end{array}\right.$$
where $\Delta (df^{b})$, $\Delta (d\omega^{b})$ is called the conversion error of accelerometer and gyroscope, they’re the error of $df^{b}$,$d\omega^{b}$. $g^{t}$, $g^{t_{m}}$ respectively indicate the magnitude of the general acceleration of gravity in the actual test area and the simulated polar area, $\omega_{ie}$ denotes the angular rate of the Earth’s rotation, $\psi^{t}$ denotes the heading angle in *t*-frame, and
(7)$$\left\{\begin{array}{l} A=\cos\lambda^{t}-\cos(\Lambda -\lambda^{t})\\ B=\sin\lambda^{t}+\sin(\Lambda -\lambda^{t}) \end{array}\right.$$

Variables *A* and *B* change slowly in a certain test period.

Considering that navigation is the main activity in the polar region, after certain simplification and approximation, Formulas (8) and (9) are obtained, which is called the marine IMU conversion error estimation formula, and its calculation accuracy is more than 90%.
(8)$$\Delta (df^{b})\approx -2\omega_{ie}\left(\begin{array}{c} A(\Delta v_{\mathrm{tE}}^{t}\sin\psi^{t}-\Delta v_{\mathrm{tN}}^{t}\cos\psi^{t})\\ A(\Delta v_{\mathrm{tE}}^{t}\cos\psi^{t}+\Delta v_{\mathrm{tN}}^{t}\sin\psi^{t})\\ -B\Delta v_{\mathrm{tN}}^{t} \end{array}\right)+(g^{t}-g^{t_{m}})\left(\begin{array}{c} \phi_{\mathrm{tE}}^{t}\sin\psi^{t}-\phi_{\mathrm{tN}}^{t}\cos\psi^{t}\\ \phi_{\mathrm{tE}}^{t}\cos\psi^{t}+\phi_{\mathrm{tN}}^{t}\sin\psi^{t}\\ 0 \end{array}\right)$$
(9)$$\Delta (d\omega^{b})\approx \omega_{ie}\left(\begin{array}{c} A\phi_{\mathrm{tE}}^{t}\sin\psi^{t}-A\phi_{\mathrm{tN}}^{t}\cos\psi^{t}+B\phi_{\mathrm{tU}}^{t}\sin\psi^{t}\\ A\phi_{\mathrm{tE}}^{t}\cos\psi^{t}+A\phi_{\mathrm{tN}}^{t}\sin\psi^{t}+B\phi_{\mathrm{tU}}^{t}\cos\psi^{t}\\ -B\phi_{\mathrm{tN}}^{t} \end{array}\right)$$

## 3. System Influence of IMU Conversion Error

After the reference error causes the IMU conversion error, it affects the simulated IMU data. In this section, it is deduced that the IMU conversion error can be equivalent to the instrument error of the simulated IMU, and the effect of different forms of error is discussed.

### 3.1. Additional Bias in Simulated IMU

Both accelerometer and gyroscope contain instrument errors, and the error model can be expressed as
(10)$$\left\{\begin{array}{c} {\tilde{f}}_{\mathrm{sf}}^{b}-f_{\mathrm{sf}}^{b}=\delta f_{\mathrm{sf}}^{b}=\delta {Κ}_{A}f_{\mathrm{sf}}^{b}+\nabla^{b}\\ {\tilde{\omega }}_{ib}^{b}-\omega_{ib}^{b}=\delta \omega^{b}=\delta {Κ}_{G}\omega^{b}+\varepsilon^{b} \end{array}\right.$$

$\nabla^{b}$, $\varepsilon^{b}$ indicates the bias of the accelerometer and the gyro, $\delta {Κ}_{A}$, $\delta {Κ}_{G}$ indicates the calibration error of the accelerometer and gyro. Taking the accelerometer as an example, let $\delta f_{\mathrm{sf}}^{b_{m}}$ be the equivalent measurement error of the polar simulated IMU, then
(11)$$\begin{array}{c} {\tilde{f}}_{ib_{m}}^{b_{m}}=f_{ib_{m}}^{b_{m}}+\delta f_{\mathrm{sf}}^{b_{m}}=(f_{ib}^{b}+\delta f_{\mathrm{sf}}^{b})+d{\tilde{f}}^{b}\\ =f_{ib}^{b}+df^{b}+\Delta (df^{b})+\delta f_{\mathrm{sf}}^{b}\\ =f_{ib_{m}}^{b_{m}}+\delta f_{\mathrm{sf}}^{b}+\Delta (df^{b}) \end{array}$$

Similarly deduced for the gyro and the ultimate result is:(12)$$\left\{\begin{array}{c} \delta f_{\mathrm{sf}}^{b_{m}}=\delta f_{\mathrm{sf}}^{b}+\Delta (df^{b})\\ \delta \omega^{b_{m}}=\delta \omega^{b}+\Delta (d\omega^{b}) \end{array}\right.$$

In the simulation test, the environmental conditions of the polar region and the working status of the INS have been simulated in the actual area, so that the IMU error characteristics of the INS under test are equivalent to those in the polar region. Taking main consideration of the IMU bias error, there is a formula that approximately holds:(13)$$\left\{\begin{array}{c} \nabla^{b_{m}}\approx \nabla^{b}+\Delta (df^{b})\\ \varepsilon^{b_{m}}\approx \varepsilon^{b}+\Delta (d\omega^{b}) \end{array}\right.$$

The formula shows that the bias of the simulated IMU contains two components: the bias of the tested actual IMU and the IMU conversion error. Therefore, the IMU conversion error is also called the additional bias of the simulated IMU.

### 3.2. Maximum Propagation Coefficient of Simulated IMU Error

To facilitate the study of estimating the specific magnitude of the additional bias and the subsequent error control and accuracy evaluation, the maximum multiple multiplied by the reference error when calculating the additional bias is referred to as the maximum propagation coefficient. Analyzing Formulas (8) and (9), when the heading vector and some reference error vectors are collinear or vertical, the additional bias is to obtain the extreme value. Table 1 shows the maximum propagation coefficient, and each variable takes the absolute value, where $\phi_{P}$, $\phi_{U}$, and Δv, respectively, indicate the magnitude of the misalignment angle of the horizontal, the up attitude and the magnitude of the velocity error, and the variable subscripts “x, y, z” represent the axial components in *b*-frame.

Considering the signs of variables A and B, the estimation formula for the maximum additional bias of the simulated IMU is:(14)$$\left\{\begin{array}{l} \;\max[\Delta (df^{b}{)}_{x,y}]\approx -2A\omega_{ie}\Delta v+(g^{t_{m}}-g^{t})\phi_{P}\\ \max[\Delta (d\omega^{b}{)}_{x,y}]\approx \omega_{ie}(B\phi_{U}-A\phi_{P})\\ \;\max[\Delta (d\omega^{b}{)}_{z}]\approx B\omega_{ie}\phi_{P} \end{array}\right.$$

### 3.3. Specific Form of Additional Bias Caused by Reference Constant and Random Bias

The reference error roughly contains system error and random error components. In order to simplify the analysis reasonably, it is assumed that the reference system error is a constant deviation, and the random error component has a complex form and is assumed to be Gaussian white noise. The specific analysis is as follows.

(1) Constant reference error may cause a constant accumulated error of the simulated IMU.

e.g., Suppose the reference speed error is a constant value of 0.1 m/s, whose vector direction is collinear with the heading vector for a long time. Meanwhile, there will be a constant component in $\Delta {(df^{b})}_{x}$, and its value is set to be about 10^−6^ g, which acts on the simulated IMU. If the actual constant bias of the IMU accelerometer to be tested on the *x*-axis is b μg, then the equivalent bias of the accelerometer *x*-axis in the constructed simulated IMU will be b + 1 μg.

(2) Random error in the reference will inevitably cause the additional random drift of the simulated IMU, which is manifested as a random walk error.

e.g., 1: Suppose that in a period, $\phi_{\mathrm{tN}}^{t}$ can be regarded as a normal distributed random variable with a standard deviation of 1’ (2.9 × 10^−4^ rad). It can be assumed that the value of $\Delta {(d\omega^{b})}_{z}$ obeys the Gaussian distribution with a standard deviation of about $7\times 10^{-5}\phi_{\mathrm{tN}}^{t}$. If the INS solving frequency *f* = 100 Hz, then the standard deviation of *z*-axis angular error accumulated in each sample is 2 × 10^−10^, constituting the bias of 2 × 10^−10^ rad/$\sqrt{0.01s}$ (about 6.9 × 10^−6^ °/$\sqrt{h}$).

e.g., 2: Assuming that the reference system has the same error as e.g., 1, it acts on the accelerometer conversion. If the carrier heading is transverse north, then the value of $\Delta {(df^{b})}_{x}$ can be set to obey Gaussian distribution with a standard deviation of $0.05\phi_{tN}^{t}$, and the standard deviation of velocity random walk accumulated for each sample is 1.45 × 10^−7^ m/s. At this time, the velocity random walk of *x*-axis is 1.45 × 10^−7^ (m/s)/$\sqrt{0.01s}$ (about 0.148 μg/$\sqrt{\mathrm{Hz}}$).

Based on the above analysis, the theoretical maximum additional random drift calculation formula is (the unit of *f* is Hz):(15)$$\begin{array}{cc} \varepsilon_{d} & =\frac{1}{\sqrt{f}}[\Delta (d\omega^{b}){]}_{\mathrm{SD}}(rad/\sqrt{s})\\ & =\frac{10800}{\pi \sqrt{f}}[\Delta (d\omega^{b}){]}_{\mathrm{SD}}({}^{\circ}/\sqrt{h}) \end{array}$$
for gyro, and
(16)$$\begin{array}{cc} \nabla_{d} & =\frac{1}{\sqrt{f}}[\Delta (df^{b}){]}_{\mathrm{SD}}(m/s^{1.5})\\ & \approx \frac{10^{5}}{\sqrt{f}}[\Delta (df^{b}){]}_{\mathrm{SD}}(\mu g/\sqrt{Hz}) \end{array}$$
for the accelerometer. “SD” means the standard deviation of IMU conversion variables.

The superposition calculation method of the random error index is different from the constant one. Assuming that the inherent random error of the tested IMU is $\sigma_{1}$, the additional random error caused by the reference error is $\sigma_{2}$, and its units are the same. It is easy to prove that the equivalent combined random error in the simulated IMU is:(17)$$\sigma_{\mathrm{gross}}=\sqrt{\sigma_{1}{}^{2}+\sigma_{2}{}^{2}}$$

## 4. Simulation Experiments and Verification

In order to verify the correctness of the theory and formula of simulated IMU additional bias, several simulation experiments were designed.

It is designed to take the simulated INS heading error as observation, and two main verifications are conducted, including experiments with a reference constant error and reference random error. First, the different influences of the same reference constant attitude error on the IMU conversion error in different carrier headings were verified quantitatively. Second, the correctness of the theories, such as the influence of INS solving frequency on additional random walks, and the random error superposition calculation in simulated IMU, was verified quantitatively.

### 4.1. Heading Calculation of 8-Orientation Simulation Test on Static Base

In the case of a constant error in the reference attitude of the *b*-frame, a simulation test of the heading solution of polar simulated INS is designed.

Setting the reference system has a pitch error Δθ of 1’ and a roll error Δγ of −2’ in a static base condition. The 8 orientations of the geographic coordinate system(*g*-frame) of the carrier were set to 0°, 45°, 90°, 135°, 180°, 225°, 270°, and 315°. The test latitude is 30° N, and the trajectory transition angle Λ=60° (to simulate the North Pole). The instrument error of the tested IMU and the initial alignment error were set to zero. Experiment duration t = 30 min, and the heading parameters calculated by the simulated INS with 8 orientations are shown in Figure 3:

The theoretical heading error results should be calculated first. Reference [16] analyzed the inertial navigation solution error of the transverse arrangement of the static base, and gave a complete expression. When it is at the North Pole, the transverse misalignment angle error of the polar INS can be expressed as:(18)$$(simulated\_INS)\phi_{t_{m}U}^{t_{m}}=-t\varepsilon_{t_{m}U}^{t_{m}}$$

That is, the upper misalignment angle error direction will diverge linearly due to the constant drift of the upper gyro.

When the carrier is level, the heading error and the upper misalignment angle error are opposite to each other:(19)$$(simulated\_INS)\Delta \psi^{t_{m}}=-\phi_{t_{m}U}^{t_{m}}$$

From Equation (9), the additional constant drift error for the *z*-axis in simulated IMU gyro is:(20)$$\Delta {\left(d\omega^{b}\right)}_{3}=\Delta (d\varepsilon_{\mathrm{tU}}^{t})\approx -B\omega_{ie}\phi_{\mathrm{tN}}^{t}$$

According to the relationship between the attitude error angle and the platform misalignment angle, the relationship between $\phi_{\mathrm{tN}}^{t}$ and the attitude error of the *b*-frame can be solved:(21)$$\phi_{\mathrm{tN}}^{t}=-\sin\psi^{t}\Delta \theta^{t}-\cos\psi^{t}\Delta \gamma^{t}$$

Combined the above formulas and noticed the relationship between the transverse heading and the geographic heading on the transverse equator ($\psi =\psi^{t}-90{}^{\circ}$), and finally:(22)$$\left(simulated\_INS)\Delta \psi^{t_{m}}\approx tB\omega_{ie}(\Delta \theta \cos\psi -\Delta \gamma \sin\psi \right)$$

According to the test latitude and trajectory transition angle Λ, “*B* = 0.866” can be calculated, substituting all the variables to obtain. The result is shown in Table 2, which is in full agreement with Figure 3.

Further experiments were conducted to verify the superimposed effect with the tested IMU’s inherent constant bias.

If the constant bias of the tested IMU “gyro drift: 0.002°/h” is consistent, referring to Equations (18) and (19), the theoretical heading error of INS after 30 min is calculated as (reference system is ideal):(23)$$\begin{array}{l} (simulated\_INS)\Delta \psi^{t_{m}}=t\varepsilon_{t_{m}U}^{t_{m}}\\ \approx 1800\times 9.7\times 10^{-9}=1.746\times 10^{-5}\approx 3.6^{\prime \prime } \end{array}$$

The constant drift setting of the inertial navigation gyroscope to 0.002°/h, and the pitch and roll error of the reference system is set as before. The theoretical heading error of the test is the sum of 3.6” and the values in Table 2.

The experiment is conducted, and the results are in accordance with the theoretical prediction.

This test actually simulates the influence of the installation error of the reference system.

### 4.2. Heading Calculation of Reference Random Error Impact on Simulated INS

A trajectory was set near (30° N, 0°). The duration was 200 s. The reference trajectory was transferred to near the North Pole to form a simulated trajectory.

(1) Verification for the calculation of the simulation IMU’s additional random drift is performed.

A random reference error was set to obey the normal distribution in the transverse north misalignment angle $\phi_{\mathrm{tN}}^{t}\sim N(0,{\left(1{}^{\circ}\right)}^{2})$, and the other errors were all set to zero. A simulation test was carried out on the three to-be-tested INS solving frequencies of 100 Hz, 200 Hz, and 1000 Hz.

The heading error was observed in the simulated INS. The results of the simulation test heading error calculation in the experiment are shown in Figure 4. The heading error value was differentially calculated in seconds, and the increment of the heading error per second is shown in Figure 5.

The heading error shows a typical random walk property (Figure 4), that is, a discrete Wiener process, and the difference is Gaussian white noise (Figure 5). Statistics on the incremental data of the heading error in Figure 4 can calculate the standard deviation of the incremental heading error per second at different frequencies.

The theoretical value was calculated first: Given the formula “$\Delta {\left(d\omega^{b}\right)}_{z}=B\omega_{ie}\phi_{\mathrm{tN}}^{t}$” holds in (9), substituting it into Equation (15) and calculating the additional random drift of the *z*-axis of the simulated gyro at the three frequencies:(24)$$\left\{\begin{array}{cc} \varepsilon_{d}(100Hz) & =\frac{1}{\sqrt{f}}[\Delta (d\omega^{b}){]}_{\mathrm{SD}}(rad/\sqrt{s})\\ & =\frac{1}{\sqrt{100}}\times 0.866\times 7.29\times 10^{-5}\times \frac{\pi }{180}(rad/\sqrt{s})\\ & =0.0227^{\prime \prime }/\sqrt{s}\\ \varepsilon_{d}(200Hz) & =0.0161^{\prime \prime }/\sqrt{s}\\ \varepsilon_{d}(1000Hz) & =0.0072^{\prime \prime }/\sqrt{s} \end{array}\right.$$

The analyzed statistics are shown in Figure 5: When *f* = 100 Hz, the standard deviation of the heading error per second increment is 0.0230 (″); when *f* = 200 Hz, it is 0.0164 (″); when *f* = 1000 Hz, it is 0.0073 (″). The theory meets the actual data.

(2) Verification for the calculation of the simulated IMU equivalent random drift combined with the reference random error.

This part is to take verifications on the simulated IMU random error superimposed effect.

Simulation 1: The reference error was set to zero, and the inherent gyro random drift of the tested INS was set to the values calculated above: $0.023^{\prime \prime }/\sqrt{s}$, $0.016^{\prime \prime }/\sqrt{s}$, and $0.007^{\prime \prime }/\sqrt{s}$. The heading error was collected in second increments and its standard deviations were counted, and the results were 0.0232, 0.0165, and 0.0073 (unit: arc seconds). The statistical data remained the same when changing into different solving frequencies.

Simulation 2: The random drift of the tested INS gyros was set as 3 × 10^−4^${}^{\circ}/\sqrt{h}$, and the random error of the transverse north misalignment angle in the reference system was set at the same time as $\phi_{\mathrm{tN}}^{t}\sim N(0,{\left(1{}^{\circ}\right)}^{2})$. The above three calculation frequencies were still set, and the simulation test was performed. According to Formula (17), the equivalent combined random drift of the simulated INS was theoretically calculated corresponding to the three solution frequencies: $0.0290^{\prime \prime }/\sqrt{s}$, $0.0241^{\prime \prime }/\sqrt{s}$, $0.0194^{\prime \prime }/\sqrt{s}$. The experiment was conducted, and the standard deviation of the simulated test heading error increment by second under the three gyro drifts were 0.0292, 0.0244, and 0.0194 (unit: arc second) on average, which meets the theoretical value.

## 5. Conclusions

This paper focuses on the evaluation and analysis of the INS polar mid-low latitude simulation test system, and it conducts an in-depth study on the quantitative analysis content of the simulated IMU conversion error. It is proved that the IMU conversion error can be approximately equivalent to the additional bias of the polar simulated IMU. Using the marine IMU conversion error estimation formula, the simulated IMU additional bias caused by different classifications of reference error is quantitatively analyzed. The following conclusions are drawn.
(1)The maximum propagation coefficient from the reference system error to simulated IMU error is derived.(2)The reference constant error has a great influence on the simulated IMU, especially the reference attitude error. Therefore, the relative installation error between the reference equipment and the tested INS should be strictly controlled.(3)The intensity of the influence of the reference random error, which can be modeled as Gaussian white noise on the simulated IMU, is related to the calculation frequency. Therefore, increasing the calculation frequency can reduce the influence of this error component. However, as far as the current IMU device accuracy level is concerned, the additional random error in the simulated IMU is relatively small.

According to the above study, it is necessary to adopt methods such as calibration, multi-system fusion, and symmetrical course planning to reduce the error of the reference installation as much as possible or to control and offset its influence on the simulated INS.

The research results of this paper can provide preliminary guidance for the selection of test reference equipment, test capability analysis, and system error evaluation of the INS polar mid-low latitude simulation test system.

## Figures and Tables

**Figure 1 sensors-22-05988-f001:**
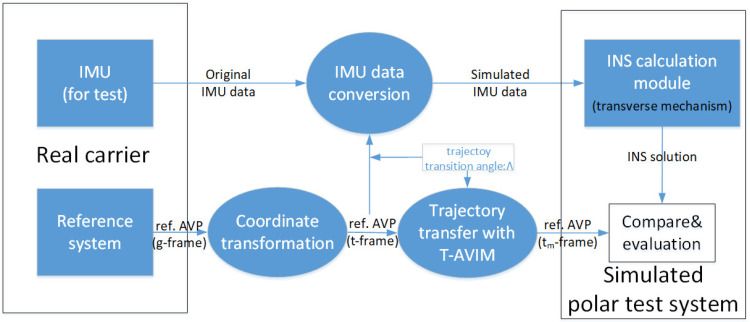
Flow chart of INS P-ML simulation test using T-AVIM under Earth sphere model.

**Figure 2 sensors-22-05988-f002:**
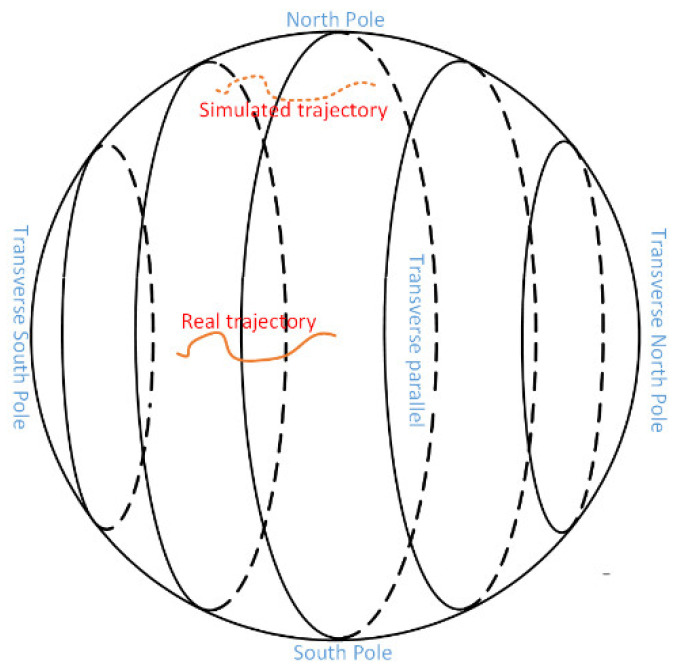
Simple demonstration of the reference trajectory transfer.

**Figure 3 sensors-22-05988-f003:**
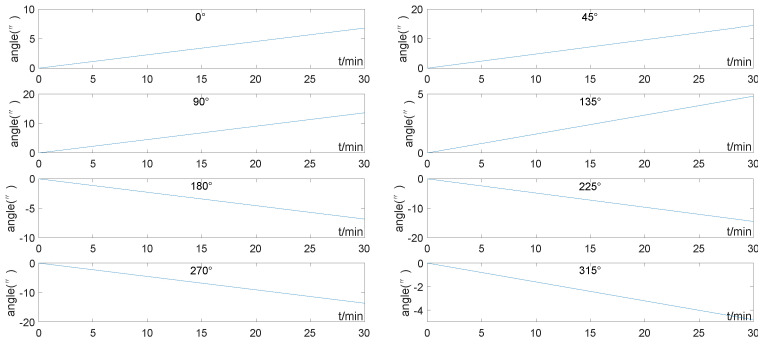
Heading error of 8-orientation simulation test on static base.

**Figure 4 sensors-22-05988-f004:**
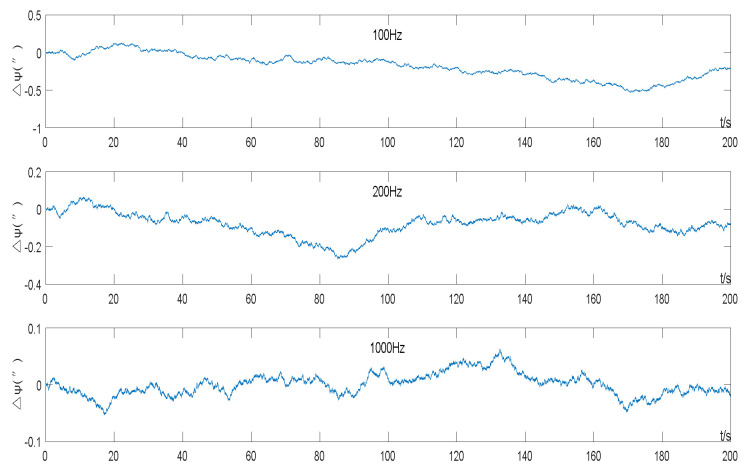
Heading error of the simulation test with different sampling frequencies.

**Figure 5 sensors-22-05988-f005:**
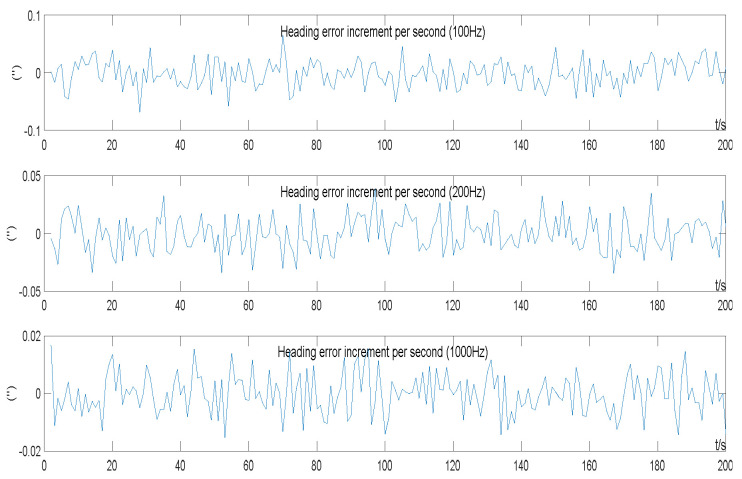
Heading error incremented by seconds in simulation test.

**Table 1 sensors-22-05988-t001:** Maximum Propagation Coefficient of Simulated IMU Error.

	Reference Error	Δv	$\phi_{P}$	$\phi_{U}$
Additional Bias				
$\Delta {(df^{b})}_{x,y}$		$2A\omega_{ie}$	$g^{t_{m}}-g^{t}$	
$\Delta {(d\omega^{b})}_{x,y}$			$A\omega_{ie}$	$B\omega_{ie}$
$\Delta {(d\omega^{b})}_{z}$			$B\omega_{ie}$	

**Table 2 sensors-22-05988-t002:** Theoretical value of 8-orientation simulation test heading error.

Carrier Heading: ψ	0°	45°	90°	135°	180°	225°	270°	315°
Simulated INS: Δψ(″)	6.82	14.46	13.64	4.82	−6.82	−14.46	−13.64	−4.82

## Data Availability

Not applicable.

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
