# Peer review of "Quantitative Study of the T-AVIM-Based Simulated IMU Error in Polar Regions"

_sensors, 2022, doi:10.3390/s22165988_

Round 1

Reviewer 1 Report

I am very glad to review this paper which is of interest for the journal. This paper is of great importance to enhance the precision of the inertial navigation system in polar regions.

This paper conducted research on the simulate IMU error and analyzed the IMU additional bias, so the problem of polar performance of INS is solved, finally the correctness of the proposed method is verified by simulation.

This manuscript can be accepted. But I think the device error may be revised as instrument error.

Author Response

Thank you for your recognition of our work, and we accept your suggestion to change the expressions.

Reviewer 2 Report

The studied topic is an interesting one, but quite abstract for the readers, especially since we are talking about a research article limited in introductory information through which to make the transition to the scientific area.

The article should be much more solid in terms of content, it is extremely succinctly presented all the information.

Recommendations:

Present in detail aspects regarding the usefulness of the solution and the novelty element that your entire research brings to the field.

Please carefully analyze the references, they are insufficient for scientific material.

The presentation of the solution is an evasive one without too much scientific importance.

Exemplify the simulation method, the algorithm developed, and the program used.

 The data presented are vague and are not presented in a chronological order that could help the reader identify the novelty or usefulness of your proposal.

The presentation of the data seems more like an enumeration of aspects and cannot be considered to have a too relevant structure. The discussions are in the second person ...

How to get a reference error caused by IMU conversion and this affecting the simulated IMU data, sounds ambiguous.

Please review the manuscript as there are some expressions that do not sound legible.

The table shows only a few data, at first glance, it seems extremely superficially treated the problem.

The table format is one that I did not find in the MDPI log template

Data interpretation is faulty.

Author Response

Thank you for your careful review, and we have made major revisions.

We are aware that the lack of overall introduction to the scheme will lead to difficulties for readers, therefore, we have supplemented the detailed introduction and its application of the scheme in Section 2 .

Some reference in relevance is added.

The simulation experiment data and ideas have been clearly rearranged and we believe it effective for verifying the correctness of proposed theory. The simulation experiment includes two parts: with reference constant error and with reference random error, and each part includes the verification of the calculation formula and the verification of the superposition theory.

We also gone through the manuscript, supplemented some formulas, and optimize many expressions of the manuscript.

Hope to get your further suggestions, thanks!

Reviewer 3 Report

It is proved that the IMU conversion error can be equivalently superimposed on the bias error of the polar simulated IMU in this manuscript. It is a valuable research result. 

Author Response

We appreciate your affirmation of our work, and we have further optimized some language expressions in the manuscript.

Reviewer 4 Report

The paper presents a quantitative analysis method of the T-AVIM based simulated IMU error in Polar Regions.The paper is reasonably organized and the test conclusion is credible.My main concerns are as follows:

1) In this paper, only the static trajectory is used to verify some errors, and it is suggested to supplement the dynamic trajectory for comprehensive verification.

2) In (13), (15), (16), (19), and (20), the ariable name is too long.

Author Response

Thanks for your careful review, and here are some issues we need to explain.

The paper is to quantitatively discuss how much reference error would cause how much simulated IMU error, if actual dynamic test is taken, to strictly verify the conclusion, the deviation of the reference parameters from the absolute benchmark (theoretical value) must be given quantitatively at every time step, it is impossible to satisfy. 

The research is to estimate instrument conversion error and used for the selection of test equipment, it is not necessary to calculate the precise error value in the actual application. The simulation test in the manuscript has entirely proved the correctness of proposed formula and we believe it effective in actual test. Thus, we are regretted not to have field test.

For variables in some formulas, we realize the name length and make modifications, but as for some special variables like heading error(Δψ),it will appear in two places, one is real reference error at mid-low latitudes, the other is the calculation parameter for polar INS. We are afraid of causing ambiguity, so we add explanations in parentheses to distinguish them, which cannot be regarded as part of the variable name.

Besides, we also optimize some expressions in the manuscript.

Hope to get your understanding.

Thanks.

Round 2

Reviewer 2 Report

We can say that this work has been constantly improved and that the research on the IMU simulation error has been analyzed and the additional IMU distortion, leading to a conclusion regarding the problem generated by the polar performance indicators of the INS. Finally, the correctness of the proposed method are checked through the simulation. The article can present some conclusions that are much better structured and of much better quality than what is found in the manuscript.

Under the given conditions, the article can be accepted.